# The Potential Modulatory Effects of Exercise on Skeletal Muscle Redox Status in Chronic Kidney Disease

**DOI:** 10.3390/ijms24076017

**Published:** 2023-03-23

**Authors:** Sara Mendes, Diogo V. Leal, Luke A. Baker, Aníbal Ferreira, Alice C. Smith, João L. Viana

**Affiliations:** 1Research Center in Sports Sciences, Health Sciences and Human Development, CIDESD, University of Maia, 4475-690 Maia, Portugal; saramendes@umaia.pt (S.M.); diogo.leal@umaia.pt (D.V.L.); 2Leicester Kidney Lifestyle Team, Department of Health Sciences, University of Leicester, Leicester LE1 7RH, UK; lab69@leicester.ac.uk (L.A.B.); alice.smith@leicester.ac.uk (A.C.S.); 3Nova Medical School, 1169-056 Lisbon, Portugal; anibalferreira@netcabo.pt; 4NephroCare Portugal SA, 1750-233 Lisbon, Portugal

**Keywords:** chronic kidney disease, skeletal muscle wasting, reactive oxygen species (ROS), oxidative stress, exercise

## Abstract

Chronic Kidney Disease (CKD) is a global health burden with high mortality and health costs. CKD patients exhibit lower cardiorespiratory and muscular fitness, strongly associated with morbidity/mortality, which is exacerbated when they reach the need for renal replacement therapies (RRT). Muscle wasting in CKD has been associated with an inflammatory/oxidative status affecting the resident cells’ microenvironment, decreasing repair capacity and leading to atrophy. Exercise may help counteracting such effects; however, the molecular mechanisms remain uncertain. Thus, trying to pinpoint and understand these mechanisms is of particular interest. This review will start with a general background about myogenesis, followed by an overview of the impact of redox imbalance as a mechanism of muscle wasting in CKD, with focus on the modulatory effect of exercise on the skeletal muscle microenvironment.

## 1. Introduction

Unlike de novo embryonic muscle formation, adult myogenesis or muscle regeneration in higher vertebrates depends on the extracellular matrix (ECM) scaffold remaining (after tissue damage), serving as a template for the muscle fibres [1]. The mechanisms of embryonic myogenesis are to some extent recapitulated during muscle regeneration (see [2,3] for a more detailed description). In brief, it is during embryonic myogenesis that the first muscle fibres are generated [4]. These are derived from mesoderm structures and are the template fibres for the following wave of additionally generated ones [5,6]. Initially, an exponential proliferation occurs up to a degree where the number of fabricated myonuclei starts decreasing, up until a steady state of synthesis rate is reached [7,8]. This leads to the establishment of a matured muscle, followed by quiescence of the progenitor cells and its occupation within the muscle fibres as satellite cells [9,10]. The myogenic rely on the satellite cells’ capacity to become activated, and to proliferate and differentiate (including self-renewal), ensuring an efficient muscle repair [11]. Satellite cells exist in a dormant state (i.e., quiescence or reversible G0 state), retaining the ability to reverse to a proliferative state in response to injury, which is essential for satellite cell pool long-term preservation [12,13,14]. Both timing and extension of satellite cells’ activation and subsequent myoblasts’ migration, in response to myotraumas to the injury sites, are partly regulated by a plethora of autocrine and paracrine factors [15,16]. These factors are released either from damaged myofibres, by the ECM or secreted by supporting inflammatory (e.g., neutrophils, macrophages) and interstitial cells, present in the niche or that migrate to the site following injury [17]. Moreover, cell-to-cell interactions are fundamental both during developmental (i.e., embryogenesis) and regenerative myogenesis [i.e., in response to physical activity (PA), trauma or disease]. These interactions allow myoblasts to adhere and fuse with myotubes during myogenesis (initial stage) [18] (Figure 1).

Satellite cells sit closely opposed to the myofibres or near capillaries, facilitating their nutrition, sitting within the ECM, which functions as a scaffold to facilitate their purpose [19,20]. Additionally, activated satellite cells undergo symmetric—give rise to two identical daughter-cells that will self-renew satellite stem cell pools—and asymmetric division—generate one stem cell and one daughter-cell committed to progress through the myogenic lineage and eventually will join the myofibre, ensuring repetitive rounds of regeneration [21,22]. These myofibres are formed by myoblast fusion, producing multinucleated myotubes, further maturing into myofibres (see [23,24] for details). Each myofibre is surrounded by a specialised basal lamina (BL)—endomysium—that harbours a specialised plasma membrane—sarcolemma—allowing neuronal signal transduction and structural stability [25,26]. The sarcolemma is anchor to the BL through transmembrane proteins—dystrophin-associated glycoprotein complex (DGC)—which allow the connection of cytoskeleton to ECM [27].

Muscle fibres are the base of skeletal muscle, being their basic contractile units [28]. These fibres are surrounded by a layer of connective tissue and are grouped in bundles [25,26]. Each myofibre is connected to a single motor neuron and expresses characteristics (e.g., molecules and metabolic enzymes) for contractile function, specifying the myofibre contractile properties, ranging from slow-contracting, fatigue-resistant/oxidative (type I) to fast-contracting, non-fatigue-resistant/glycolytic (type II) fibres. Moreover, the proportion of each fibre type determines overall contractile property within the muscle [29]. The connective tissue that surrounds the skeletal muscle functions as a framework, combining myofibres with myotendinous junctions (i.e., the place where myofibres attach to the skeleton), transforming myofibre contraction into movement [30]. Hence, the skeletal muscle functional properties are dependent on myofibres, motor neurons, blood vessels and ECM.

Skeletal muscle maintenance is accomplished by an interplay between multiple signalling pathways, including two major ones that control protein synthesis, IGF1-PI3K-Akt-mTOR pathway (positive regulator) and myostatin-Smad2/3 pathway (negative regulator) [3,31]. These interconnected pathways control and coordinate hypertrophic and atrophic signalling, creating a balance between protein synthesis and proteolysis [32,33].

Skeletal muscle cells are not isolated elements; they are inserted in their ecological niche, creating a social network with their surroundings. This context consists of interstitial cells, vascular features, ECM proteins and soluble factors, which together constitute the skeletal muscle microenvironment [34]. This microenvironment must be adequate to support skeletal muscle functions and allow a suitable regeneration after assault, such as that imposed by disease (discussed in Section 3, Section 4 and Section 6) or exercise (discussed in Section 4, Section 5 and Section 6). When this does not occur, we may be confronted with processes leading to muscle wasting.

## 2. REDOX Imbalance as a Mechanism of Muscle Wasting

Skeletal muscle atrophy is a process that occurs as a result of conditions such as disuse, malnutrition, aging and in certain states of disease. Nonetheless, it is characterized firstly by a decrease in muscle mass (and volume), force production and, on a more detailed perspective, by a diminishment of protein content and fibre diameter [35]. Moreover, the primary loss in muscle strength that occurs with atrophy results from the rapid destruction of myofibrils, the contractile machinery of the muscle, constituting around >70% of the muscle protein [36].

Among all the potential aetiological foundations of muscle wasting, reactive oxygen species (ROS) generation, including the oxidative damage and/or the defective redox signalling, has stood out as the possible main explanation [37,38,39].

ROS are reactive molecules that contain oxygen, and this family is comprised of free radicals (i.e., species with at least one unpaired electron) and nonradical oxidants (i.e., species with their electronic ground state complete). The chemical reactivity of the various ROS molecules is vastly different; for instance, hydroxyl (●OH), the most unstable, reacts immediately upon formation with biomolecules in its vicinities, whereas hydrogen peroxide (H_2_O_2_) is capable of crossing cell membranes to exert its effects beyond its original compartment [40,41,42] (Table 1).

ROS are generated by various sources, mainly endogenous sources, including mitochondrial respiratory chain enzyme, nicotinamide adenine dinucleotide phosphate oxidase (NOX) activity, microsomal cytochrome P450 and xanthine oxidase; and exogenous sources such as ultraviolet radiation, X- and gamma (γ)-rays, ultrasounds, pesticides, herbicides, and xenobiotics [43]. Superoxide anion (O_2_-^●^) is the most frequently generated radical, under physiological conditions. Its main source is the inner mitochondrial membrane, in the complexes I and III, during respiratory chain, by the inevitable electron leakage to O_2_ [44,45]. It can also be generated in the short transport chain of endoplasmic reticulum upon electron leakage and during NOX activity, by transferring one electron from nicotinamide adenine dinucleotide phosphate (NADPH) to O_2_ [46].

To cope with ROS, the cells have developed control systems to regulate oxidation/reduction balance, since redox balance is critical. A key component is the antioxidant system, which prevents ROS accumulation and deleterious actions. The cells contain both enzymatic and non-enzymatic antioxidants that work by mitigating ROS effects and by drastically delaying/preventing oxidation from happening. Key enzymatic antioxidants are superoxide dismutase (SOD), catalase, glutathione peroxidase (GPx) and thioredoxin (Trx), whereas non-enzymatic are mainly vitamin C (ascorbic acid) and E (tocopherol), zinc and selenium, glutathione, plant polyphenols and carotenoids [47,48]. These act primarily by using three different strategies: (1) scavenging ROS; (2) converting ROS molecules into less reactive ones, and (3) chelation via metal binding proteins. Throughout the cells, antioxidants are compartmentalized in both organelles and cytoplasm, but also exist in the interstitial fluid and blood [49].

ROS are normal products of cell metabolism with significant physiological roles. They regulate signalling pathways (redox signalling) by changing the activity of structural proteins, transcription factors, membrane receptors, ion channels and protein kinases/phosphatases [50,51]. ROS physiological roles depend partly on antioxidant control, establishing a redox balance. When redox homeostasis is disrupted, due to the rising of ROS levels and the unlikely neutralization by the antioxidant defence, a state referred to as oxidative stress (OS) occurs. This leads to an impairment of redox signalling and induces molecular damage to biomolecules [52,53]. Moreover, OS has a graded response, with minor or moderated changes provoking an adaptive response and homeostasis restoration, whereas violent perturbations lead to pathological insults, damage beyond repair and may even lead to cell death [53]. Interestingly, something that is not appreciated often is that our understanding of “low” or “high” response regarding ROS levels is somewhat imprecise, redox time-courses in vivo are scarce and our knowledge is based of immunohistochemical analysis or measuring more stable elements of the family [54,55].

As in other tissues, redox signalling in skeletal muscle has important roles, being the base of skeletal muscle function to elicit exercise adaptation. It supports the neuromuscular development and the long-term remodelling/adaptation of contractile activity [56,57]. Moreover, regulated ROS levels are also involved in skeletal muscle regeneration, regulating the activity of skeletal muscle stem cells, through redox-sensitive signalling pathways [58] (Figure 2).

When an ROS overproduction occurs, cells are capable of maintaining a redox state by activating distinct transcription factors that induce the transcription of antioxidant enzymes to tilt the balance back to homeostasis, protecting them from OS [59,60]. One important transcription factor is the nuclear factor erythroid 2-related factor 2 (Nrf2), which is a ubiquitous protein that modulates OS [61]. In response to elevated ROS levels, Nrf2 triggers the expression of NADPH quinone oxidoreductase (NQO1), heme oxygenease-1 (HO-1), glutamate-cysteine ligase catalytic (GCLC) and glutamate-cysteine ligase modifier (GCLM), which are enzymes involved in redox homeostasis maintenance, cellular defence and detoxification [62,63]. Moreover, enzymes that encapsulate the redox cycling group, mediating the elimination of ROS such as thioredeoxin, thioredoxin reductase, sulfiredoxin, peroxiredoxin, gluthatione peroxidase, superoxide dismutase 1 (SOD1), catalase and various glutathione S-transferases, are all of them targeted by Nrf2 [64].

However, during ageing, cells produce even more ROS, mainly from mitochondria and NOX, and even though the activity of antioxidant enzymes in cells and muscle also increases with age, this compensatory adaptation is not sufficient to neutralize ROS levels [37,38,39]. These increased ROS levels cause deleterious macromolecules oxidative modification, leading not only to various cellular dysfunctions, but also affecting signal transduction pathways that control multiple essential cellular processes, such as protein turnover, mitochondrial homeostasis, energy metabolism, antioxidant gene expression and redox balance (see, for example, [65] for more details). Moreover, the systemic increase in ROS, associated with an OS state, increases proinflammatory transcription factors levels, for instance, nuclear factor kappa B (NF-kB) [66,67]. NF-kB regulates specific UPS genes and leads to the expression of proinflammatory cytokines such as IL-6 and TNF-α that are involved in the development of muscle atrophy [68,69,70].

In summary, ROS load increment and the establishment of an OS state are detrimental to muscle function and are associated with the mechanism of skeletal muscle atrophy [71].

There are two common but distinct conditions that are characterized by skeletal muscle loss, which are sarcopenia and cachexia. In sarcopenia, skeletal muscle loss occurs in a slow and progressive way, being associated with ageing process (in the absence of disease), whereas, in cachexia, skeletal muscle loss is associated with inflammatory conditions (e.g., AIDS and sepsis) and chronic diseases such as cancer, diabetes, obesity, chronic obstructive pulmonary disease, chronic heart failure, chronic liver disease and chronic kidney disease [72,73,74].

## 3. REDOX Imbalance in CKD

CKD consists of a progressive and irreversible loss of kidney function in that, in the more advanced stages of the disease, patients require renal replacement therapy or renal transplantation [75]. The aetiologic factors of the myopathy observed in CKD patients are diverse, from the kidney disease itself, regardless of the need for renal replacement therapy, to the actual dialysis treatment and the typical chronic low-grade inflammation [76,77]. The skeletal muscle fibres of CKD patients present several abnormalities, such as changes in the capillarity, contractile proteins and enzymes [78]. In dialytic patients, this occurs to a greater extent to those who do not undergo dialysis, where atrophy is normally particularly observed in type II fibres [78]. This can be partially explained by the substantial amino acid loss during dialysis, a reduced energy and protein intake and low PA levels, which are recognised to be even lower on dialysis days [79,80,81]. In fact, these patients present a catabolic environment due to a dysregulated state of energy and protein balance, which includes altered muscle protein metabolism—increased protein degradation (e.g., activation of ubiquitin–proteasome system) (more noticeable) and decreased protein synthesis (e.g., suppressed IGF-1 signalling) (less observed)—and impaired muscle regeneration—satellite cell dysfunction [82]. Furthermore, the haemodialysis procedure itself can stimulate protein degradation and reduce protein synthesis, persisting for 2 h after dialysis [83]. Moreover, even though increasing protein intake (and calories) could enhance protein turnover, the haemodialysis responses were not fully corrected [84,85,86]. CKD has been previously described as a model of ‘premature’ or ‘accelerated’ ageing, associated with a redox imbalance. However, since the mechanisms of age-related muscle loss are similar, but not the same as the CKD-induced, it may be proposed that the two-simile combined amplifies the dysregulated mechanisms [87,88] (Figure 3).

Skeletal muscle wasting appears to be a shared feature in the presence of disease, which implies that disease itself can trigger a muscle atrophic response, suggesting that skeletal muscle acts as a source of amino acids providing nourishment for other tissues [89,90,91].

The dysregulation of skeletal muscle function observed in CKD may also be caused by the presence of uremic toxins, which are normally filtered and excreted by healthy kidneys. However, when kidney function is impaired or inexistent, as in CKD, these uremic toxins are accumulated in the circulation and target other tissues [92,93]. Haemodialysis is in some cases incapable of removing uremic toxins such as protein-bound toxins [i.e., indoxyk sulfate (IS) and p-cresyl sulfate] due to their high affinity to serum albumin [94,95]. The accumulation of these uremic toxins appears to exert negative effects on myoblast proliferation and myotube size (in vitro), skeletal mass (in vivo), reduction of instantaneous muscle strength (loss of fast-twitch myofibres; in vivo) and is accompanied by intramuscular ROS generation [96,97,98]. High levels of ROS induce the expression of inflammatory cytokines by the muscle, such as tumour necrosis factor (TNF)-α [99,100]. This increase in TNF-α stimulates myostatin expression via NF-kB pathway, which further stimulates myostatin expression accompanied by a rise in IL-6 release [101]. As a result, these activated pathways further increase ROS production by NADPH oxidase [99]. These inflammatory cytokines are known to be elevated in CKD patients, alongside a more pronounced myostatin expression [101,102].

Local high levels of ROS and the subsequent cascade of events (i.e., decreased antioxidant defences and increased inflammatory response) [103] disturb ECM synthesis/degradation homeostasis, favouring excessive collagen deposition, thus promoting tissue fibrosis [104,105]. Additionally, in these more severe CKD stages, skeletal muscle satellite cells and myoblasts are surrounded by an altered microenvironment composed of fibrotic tissue, fat and inflammatory cells [106,107]. The imbalanced crosstalk between resident cells and ECM in the skeletal muscle of CKD patients leads to the production of numerous growth factors, proteolytic enzymes, angiogenic and fibrogenic factors [108,109]. Interestingly, a study by Dong and colleagues [110] observed a differentiation effect of myostatin on fibro-adipogenic progenitors (FAPs), being that myostatin stimulated the proliferation and differentiation of FAPs isolated from EGFP-transgenic mice, leading to fibrosis in the skeletal muscle of CKD mice. An increased α-smooth muscle actin expression was also observed, with the in vivo inhibition of myostatin suppressing both CKD-induced FAP proliferation and muscle fibrosis. This provides a foundation for elucidating what the mechanisms of fibrosis may be in human CKD patients. In a nutshell, these patients present high levels of ROS that increase TNF- α, which stimulates muscle myostatin production. This consequently leads to FAPs proliferation and differentiation, further stimulating muscle fibrosis.

The net consequence of these alterations firstly involves the satellite cell population exhaustion (i.e., loss of activity) or decreased capacity to mediate repair over time, progressively leading to atrophy and loss of individual muscle fibres, associated with concomitant loss of motor units [111]. In fact, it has been already reported that a fibrotic state-derived excess ECM accumulation has a negative impact on muscle force production, thus suggesting that ECM alterations can have significant functional repercussions, with current research highlighting the ECM-cellular interactions as key to better understanding it [112,113]. Keeping this in mind, it has been reported that human-derived muscle cells isolated from CKD patients display and retain CKD-specific cachexia phenotypes in vivo outside of their microenvironment [114]. In addition, there is a reduction in certain muscle properties related to its overall metabolic function (i.e., muscle quality) due to fat infiltration and other non-contractile material [115]. This decrease in overall muscle architecture results in an increased susceptibility to mechanical stress and muscle fibre necrosis. Hence, it is important that ECM microenvironment be actively remodelled to allow ECM cleavage fragments to be released. These “cleaning” programs are activated by endothelial cells sensing mechanical forces such as the ones produced during physical exercise [116,117].

CKD development profoundly linked to OS, in which Nrf2 inactivation seems to be essential. Interestingly, CKD patients appear to have balance between Nrf2 and NF-kB expression; conversely, in CKD patients, under haemodialysis, it has been observed that an Nrf2 expression downregulation was accompanied by NF-kB upregulation [118,119]. Since Nrf2 downregulation contributes to OS and inflammation, it plays a role in causing cardiovascular disease and other complications in CKD patients [120]. Moreover, low levels of Nrf2 increase fibrosis markers, with fibrosis being observed in several tissues in CKD patients, such as kidney, skeletal muscle and heart [121,122,123].

Additionally, CKD has also been associated with patients with physical inactivity, which is linked with adverse clinical outcomes, increased risk of morbidity and mortality [124].

## 4. Exercise in Chronic Kidney Disease

Haemodialytic CKD patients are considerably less physically active than their age-matched counterparts [125,126]. Additionally, despite the diverse aetiologic factors of muscle wasting and decreased muscle quality observed in CKD patients, physical inactivity has been proposed as one of the major contributors [127,128,129]. In fact, a study performed on CKD patients showed similarly low levels of PA between two groups of CKD patients separated depending on disease severity [pre-dialytic (stage 3–4) vs. haemodialytic patients (stage 5)] [130].

Physical inactivity along with the disease itself leads to the patients experiencing skeletal muscle wasting, contributing to frailty, and limiting exercise tolerance [126]. Moreover, considering that CKD patients experience anaemia, hypertension, bone loss and take medications, it is understandable that these patients avoid exercise [131]. This result is unlucky since exercise is beneficial for cardiovascular health and helps with slowing down the progressive skeletal muscle mass loss [132]. Additionally, living a sedentary life and suffering from muscle mass loss negatively affect health [132]. For instance, a single resistance exercise session has been shown to be able to stimulate protein anabolism in haemodialytic patients [133], and 21 weeks of endurance exercise was found to improve protein metabolism markers (e.g., IGF-1 and myostatin). Although it may be demanding of dialysis patients to engage in moderate to vigorous exercise sessions, those who can often experience great benefits. Strategies such as regular resistance and aerobic exercise have shown promising effects in reducing the progression of sarcopenia [102,134,135,136,137]. In short, resistance training has shown to be effective in improving skeletal muscle strength and functional capacity and stimulating muscle hypertrophy (e.g., increase in type I, type IIa and type IIx muscle fibre cross sectional areas) [138,139,140,141,142], whereas aerobic training appears to significantly increase aerobic capacity and exercise duration, reduce intra- and interdialytic systolic and diastolic blood pressure, diminish arterial stiffness, increase dialysis efficiency, enhance exercise-induced capillarization in the muscle, improve quality of life (reducing anxiety symptoms), and even exert comparable effects with those of resistance training (i.e., muscle strength) due to poor initial physical state of patients [143,144,145,146,147,148,149,150,151].

Moreover, CKD is associated with a dysregulated myokine activity and a systemic increase in cytokines [152,153,154]. In response to exercise, skeletal muscle releases myokines (e.g., IL-15 and IL-6), which exert positive physiological effects on skeletal muscle and bone [155]. This crosstalk through the skeletal muscle secretome (e.g., IGF-1 and myostatin) positively influences bone health [155,156,157]. In CKD, intradialytic resistance training showed an elevation in osteoprotegerin, which acts by avoiding/protecting excessive bone resorption [158]; bone-specific alkaline phosphatase, another bone resorption inhibitor, showed elevation in resting concentrations after an 8-week intradialytic resistance exercise [159]. For more detailed information about skeletal muscle and bone crosstalk in CKD, see [160].

Connective tissue accumulation (e.g., ECM) has been observed in aged skeletal muscle [161]. A study with aged rats submitted to a resistance exercise protocol—3 times a week for 12 weeks—has shown that training mitigated the age-associated increase of connective tissue. These results can be extrapolated to CKD, since fibrosis is also present in this population [110].

In sum, although there is extensive evidence of the benefits of exercise in CKD, studies showing exercise-induced mechanistic ROS modulation are still lacking.

## 5. The Impact of Exercise in the REDOX System

Exercise puts pressure on body structures and organs, so blood must be delivered in quantity to the skeletal muscle, heart, lung (among others) rich in oxygen and nutrients to atone for that [162]. However, this stressor leads to an oxygen supply insufficient for the demands of the body, and then, in response to that, many tissues produce ROS [163]. Under normal and healthy conditions, with oxidative levels within a normal range, the available free radicals promote vasodilatation, production of muscle force and maintenance of its content, signal transduction and other related activities [58,164]. In the muscle, contractions during exercise also induce ROS formation, with this upregulating the activity of transcription factors such as NF-kB, activator protein 1 (AP-1) and NRF2, which leads to a more pronounced activity of antioxidants enzymes, inducing muscle adaptations and protecting it from periods of increased OS [165,166,167]. A study performed in old rats who performed 12 weeks of treadmill-run exercise observed an increased Nrf2 expression [168]. Moreover, a study performed in recreationally active males observed an exercise-induced Nrf2 elevation to 3 h of eccentric contractions of the knee extensors [169].

On this basis, exercise has been shown to enhance ROS detoxifying pathways by increasing the activity of SOD, Gpx, catalase and the master regulator of antioxidant defence, Nrf2 [170,171]. It is the upregulation of these detoxifying pathways that appears to be essential for the adaptive protection developed to work against detrimental effects of OS [172]. For instance, the sarcoplasmic reticulum, which releases Ca2+ necessary for muscle contraction, is highly sensitive to ROS levels, with dysregulated increments in ROS reducing myofibrils sensitivity and therefore affecting muscle contraction [173,174]. Another example that corroborates that ROS effects are dependent on their levels is observed when talking about JNK/SMAD signalling axis, responsible for muscle growth via SMAD2 phosphorylation leading to myostatin inhibition [175]. Low levels of ROS induce JNK phosphorylation, followed by SMAD2 phosphorylation and consequently muscle growth (transient activation of JNK), whereas high levels of ROS also activate JNK but deactivate phosphates, resulting in JNK persistent activation, and were associated with muscle adaptation failure [51,175]. Excess of free radicals, due to intensive exercise or not, may result in OS, putting molecules (i.e., protein, lipids and DNA) at risk for oxidative modifications [53,100]. Proteins are the most susceptible to oxidative modifications, with the more common type of oxidation modification being carbonylation, altering protein conformation leading to partial or total inactivation [176]. The direct consequence is loss of function or structural integrity having wide downstream effects leading to cell dysfunction [177]. PA appears to promote protection against protein carbonylation, which may occur due to antioxidant defence activation or increased protein carbonyls turnover [178]. Other types of oxidation modification that proteins are susceptible to are, for example, tyrosine nitration, S- glutathionylation and advanced glycation end products (AGEs) (see [179,180] for more detailed description of these processes).

Beneficial changes observed in muscle occurs in response to long-term, regular, and moderate training due to muscle adaptation, whereas acute and strenuous exercise provokes excessive free radicals, causing OS damage and fatigue and impacting the body’s health and exercise capacity [181,182]. Moreover, exercise modulation through ROS towards muscle provokes different effects on structure and function; this is majorly dependent on the type of training, which leads to activation of different pathways. In general, exercise is divided into two groups: aerobic/endurance exercise and resistance exercise. In endurance (non-exhaustive) training, the source of energy is mainly from the mitochondrial biogenesis, dependent on ROS production by exercise, modulated by peroxisome proliferator-activated receptor gamma coactivator 1-alpha (PGC-1α), the principal pathway to rise oxidative capacity of the muscle [183,184]. Regarding resistance training, the produced ROS activates signalling pathways such as IGF-1 and PI3K/AKT/mTOR, and they are associated with increments in protein synthesis [185]. Additionally, in sprinting, a short-term anaerobic exercise, high levels of ROS are produced mainly by NOXs and xanthine oxidase system; in this case, ROS production by mitochondria is less noticeable [186,187]. Moreover, in general, both resistance and endurance (exhaustive) training are shown to increase ROS levels by the skeletal muscle leading to OS, an increase in cortisol levels and a transitory immunosuppression [39]. In short, together aerobic and resistance training reduces OS, increasing resistance against it, and improves antioxidant status in the long term [188,189,190,191,192,193,194,195,196,197,198].

Finally, it appears that the influence that exercise has on the metabolism and on the redox system may explain the already proven benefits of exercise in health and disease.

## 6. The Potential Modulatory Effects of Exercise on Skeletal Muscle Redox Status in CKD

It has been already established that exercise is the main stressor that drives skeletal muscle remodelling and metabolic adaptation, and that it achieves that by, in a simple way, stressing the body to produce free radicals and at the same time stimulating it to generate antioxidants to maintain homeostasis, a new homeostasis, being more prepared for the next stress, adapted. However, CKD patients experience elevated OS, and the increase in free radicals induced by acute exercise, especially in unaccustomed patients, could further shift the imbalanced redox status to an even more pro-oxidant state, impairing skeletal muscle metabolism [199]. In response to that, our group has already shown that unaccustomed exercise creates a large inflammatory response in the muscle and that expression of inflammatory cytokines such as IL-6, MCP-1 and TNF-α was upregulated [136]. Still, this is no longer present after a period of training, showing that exercise does not appear to elicit an ongoing and detrimental inflammatory response in the muscle, but an adaptive response instead [136]. Similar to unaccustomed exercise in CKD patients, it can be partially observed with the incidence of the overtraining syndrome (OTS), in which a state of chronic OS is observed due to intensified training/competition and inadequate post-exercise/competition recovery, leading to a persistent fatigue and decline in physical performance [200]. Moreover, a study from our group showed that, after intensified training, leukocyte phagocytic activity decreases and testosterone levels were blunted, showing dysfunction of inflammatory response and at the hypothalamic-pituitary gonadal axis [201]. Interestingly, in OTS are also observed OS blood markers, for example, persistence for more than a month of a reduced glutathione depletion after an ultra-endurance marathon [202]. In these cases, resorting to an antioxidant treatment has been shown to be helpful in restoring muscle weakness and force production [203,204]. In CKD, a systematic review on the use of antioxidants in CKD patients (pre and post dialysis) shows that, in predialysis patients, it may help go prevent end-stage kidney disease, but more powered studies are needed to assess this finding [205].

CKD is considered by some a form of accelerated ageing, so we can withdraw data from older adults. For instance, after 12 weeks of moderate resistance training in elder people, it was observed that ROS generation and OS were decreased [197]. Another study also showed similar results: increase in muscle strength and function associated with decrease in OS markers and enhanced mitochondrial functions [206]. However, one study demonstrated no significant changes in OS biomarkers after aerobic exercise [196]. In sum, it appears that exercise has a positive role in elderly people, with them having OS levels similar to untrained young subjects when exercising [207]. Therefore, it is speculated that decrease in ROS generation, and consequently OS reduction, which could be accompanied by increase in muscle strength and function, may be observed in CKD, despite further evidence still being required. Moreover, like the inflammatory response observed in CKD, aged muscle produces high levels of ROS after acute exercise, while chronic exercise prepares and protects muscle against oxidative damage [208]. In CKD, the majority of studies report disease functional parameters’ improvement after a period of training but left out reports about OS markers or investigate mechanisms that cause the exercise benefits observed. A 6-month study performed on haemodialysis patients separated into two groups, intradialytic training (bedside cycling) or no-exercise control group, observed a chronic reduction in various redox status parameters, such as protein carbonylation and lipid oxidation, and an increase in enzymes responsible for ROS detoxification such as catalase and glutathione as an effect of regular exercise [209]. Additionally, this was also accompanied by an increase in aerobic and functional capacity, observed by an elevated peak oxygen consumption, and improved scores on the North Staffordshire Royal Infirmary (NSRI) walk test, and on the 60-s sit-to-stand (STS-60) test [209]. A 4-month intradialytic exercise training (cycling) could reduce plasma lipid peroxidation [210]. The same was observed after 12 weeks of aquatic exercise [211]. Interestingly, a study compared OS parameters in untrained volunteers, CKD patients and professional athletes before and after a strenuous exercise in a rowing cycle ergometer and showed that only athletes presented elevation of antioxidant enzymes due to limited antioxidant capacity in both untrained and dialysis patients, yet the last exhibited increased OS [212]. Moreover, resistance exercises during dialysis appear to be capable of inducing Nrf2 activation [213] (Figure 4).

## 7. Final Remarks

The balance between muscle mass synthesis/breakdown is essential for the normal function of the muscle, which is partly regulated by ROS. More research is accumulating regarding the impact of redox imbalance in the process of muscle wasting, even though the exact mechanisms are still to be determined. Moreover, the potential influence of exercise on the attenuation of muscle wasting in CKD patients appears to be gaining points. However, it is urgent that worldwide exercise programs be implemented to better solidify the existing results to date. Although numerous dialysis patients may appear too frail and incapable of engaging in exercise sessions, those who do it have experienced the benefits [214]. In these cases, less vigorous exercise offers value, and these types of adaptations will help to gradually lessen some clinicians’ misconceptions of exercise as a potential contraindication to the patients’ health [214]. Furthermore, besides the compelling evidence of the health benefits, there may also be impactful advantages to the healthcare systems, by reducing collateral costs of CKD patients, such as interventions associated with disease complications. More and more, we believe that the cost savings in the long-term probably overcome the financial limitations that are sometimes still imposed and impede the introduction of exercise programmes as routine in clinical units. Since CKD patients who undergo dialysis experience inevitable sedentary time during treatment, we encourage the implementation of intradialytic exercise interventions as a coadjutant therapeutic strategy to reduce or at least decelerate CKD-associated muscle wasting.

## Figures and Tables

**Figure 1 ijms-24-06017-f001:**
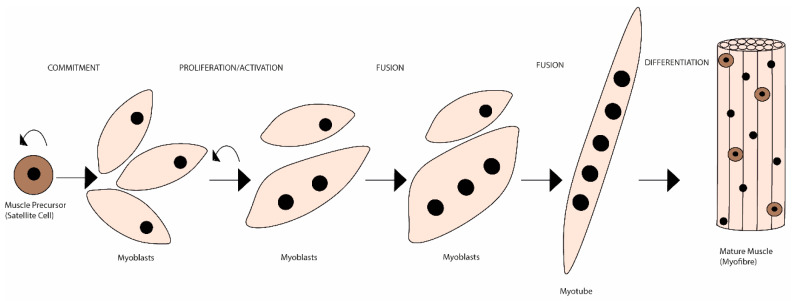
Schematic representation of the mammalian skeletal myogenesis process. Upon muscle injury, a resident population of quiescent skeletal muscle satellite cells can become activated, start to proliferate and differentiate into myoblasts. Over the course of several days, these myoblasts fuse together to form multinucleated myotubes. Further, myoblasts can also fuse to the already existing myotubes to create even larger myotubes, which will eventually align to form muscle fibres. This whole process is regulated by many internal and external cues.

**Figure 2 ijms-24-06017-f002:**
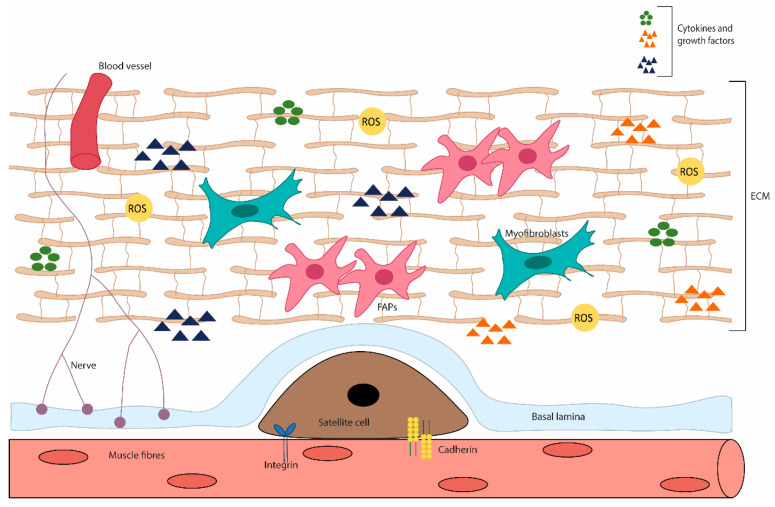
Diagram of the skeletal muscle microenvironment. This niche is composed of various cell types and ECM proteins. In adult skeletal muscle, the quiescent satellite cells stand on the myofiber, under the basal lamina, being surrounded by the ECM, containing blood vessels, nerves, immune cells, fibro-adipogenic progenitors (FAPs), adipocytes and myofibroblast. The satellite cell states are regulated by their interactions with the surrounding microenvironment, direct interaction (e.g., M-cadherin) between muscle fibres and satellite cells; or interact with a variety of components of the ECM and cytokines and growth factors. In addition, stromal cells present can physically interact with satellite cells and release cytokines, growth factors and ECM components, which influence the behaviour of satellite cells, contributing to muscle growth, homeostasis and regeneration.

**Figure 3 ijms-24-06017-f003:**
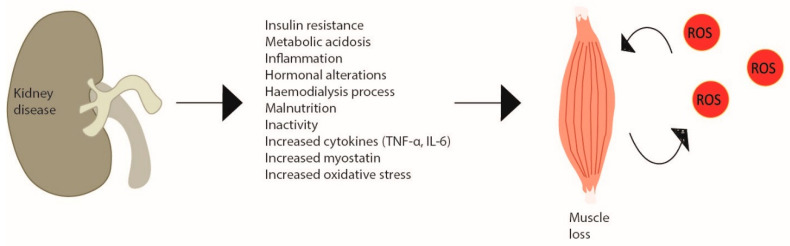
Skeletal muscle wasting induced by chronic kidney disease. Chronic kidney disease creates metabolic changes due to inflammation, haemodialysis increased cytokine production and myostatin and especially oxidative stress, which leads to skeletal muscle atrophy inducing a catabolic program and a vicious cycle of ROS production in site. In CKD patients, this is observed by decreased muscle strength and increased weakness.

**Figure 4 ijms-24-06017-f004:**
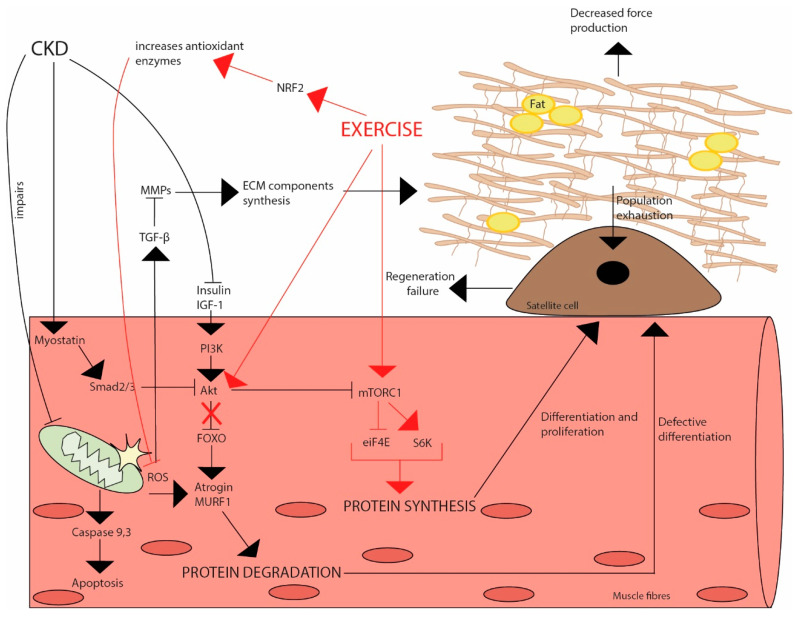
Factors affecting skeletal muscle maintenance in CKD patients. In CKD patients, various factors interact and consequently transduce their effects intracellularly, which affects skeletal muscle maintenance. For instance, insulin and IGF-I positively regulate skeletal muscle due to the activation of mTORC1 through PI3k/Akt, initiating protein synthesis. Later, stimulation of PI3k/Akt by insulin increases FOXO phosphorylation, activates MuRF1 transcription and Atrogin-1, leading to protein degradation via ubiquitin–proteasome pathway. On the other side, myostatin, a negative regulator, leads to SMAD2/3 phosphorylation, reduces Akt activation and consequently FOXO phosphorylation, which inhibits its translocation to nucleus, again accelerating protein degradation. Defective mitochondrial function, due to kidney damage, increases local ROS production, resulting in muscle protein degradation through activation of MuRF1 and Atrogin-1 transcription. These defective mitochondria lead to activation of caspase 9,3 triggering intrinsic apoptotic pathway. Transforming growth factor-β (TGF-β) is activated due to the increased exposure to ROS, while, among other functions, acting as ECM preservatory. It enhances matrix protein synthesis and suppresses ECM degradation proteins such as matrix metalloproteins, which happens in CKD due to an exaggerated activation of TGF-β. Extravagant ECM production leads to fibrosis, impinging on muscle quality, decreasing its force production. This microenvironment affects satellite cells, leading to population exhaustion and regeneration failure; protein degradation also leads to defective differentiation. On the other hand, exercise activates mTORC1, mediating S6k activation, thus promoting protein synthesis and differentiation/proliferation of satellite cells. Additionally, exercise can increase Akt activation, consequently translocating FOXO to the nucleus, blocking protein degradation pathways. Moreover, exercise elicits an increased Nrf2 expression, leading to an elevation in the expression of antioxidant enzymes, therefore decreasing ROS levels.

**Table 1 ijms-24-06017-t001:** The most common reactive oxygen species, antioxidants and respective scavenging reactions.

Reactive Oxygen Species-Oxidants	Antioxidants	Enzymatic Scavenging Reactions
Superoxide radical (O_2-_^●^) (Rad)	Superoxide dismutase (Enz), Vit C (Non-Enz)	2O_2_-^●^ + 2H+ -> O_2_ + H_2_O_2_
Hydrogen peroxide (H_2_O_2_) (Non-Rad)	Catalase (Enz), Glutatione peroxidase (Enz)	2H_2_O_2_ -> 2H_2_O + O_2_ H_2_O_2_ + 2GSH -> 2H_2_O + GSSG
Hydroxyl radical (OH^●^) (Rad)	Glutatione peroxidase (Enz), Vit C (Non-Enz)	GSH + OH^●^ -> GS^●^ + H_2_O

## Data Availability

Not applicable.

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
