# Peer review of "The Potential Modulatory Effects of Exercise on Skeletal Muscle Redox Status in Chronic Kidney Disease"

_ijms, 2023, doi:10.3390/ijms24076017_

Round 1
Reviewer 1 Report
Manuscript ID: ijms-2253996
Title: "The potential modulatory effects of exercise on skeletal muscle redox status in Chronic Kidney Disease"
In this review, Mendes et al. provide a comprehensive and informative review of the potential modulatory effects of exercise on skeletal muscle redox status in Chronic Kidney Disease. Broadly speaking, the manuscript is very clear and very informative for muscle redox status in Chronic Kidney Disease.
Comment: Here are comments that are more specific:
Introduction:
In the first paragraph of the Introduction, the authors gave emphasis on myogenesis, Is there any relation between exercise on skeletal muscle redox status in Chronic Kidney Disease, better to simplify this in lieu of the title.
The mechanisms of embryonic myogenesis are to some extent recapitulated during muscle regeneration (see [2] and [3] for a more detailed description).
Provide some brief description of embryonic myogenesis, so that readers can understand here
“The dysregulation of skeletal muscle function observed in CKD may also be caused by the presence of uremic toxins, which are normally filtered and excreted by healthy kidneys. However, when kidney function is impaired or inexistent, as in CKD, these uremic toxins are accumulated in the circulation and target other tissues…”
Provide the references on it
I found several places, where the authors need to provide references.
Regarding the Figures, I am not sure whether the authors need copyright or not, need to confirm.
Kindly provide the future prospects in this direction

Author Response
Thank you very much for the the thorough review. Please see our detailed responses in the attached document.
Kindest regards,
The Authors

Reviewer 2 Report
The manuscript looks great. Authors have done great work for scientific integrity which highlights molecular mechanisms impacting CKD.
Author Response

(The authors gave the same response as above.)

Reviewer 3 Report
The study is devoted to the consideration of the effects of exercise on skeletal muscle redox status in Chronic Kidney Disease. In this review, the authors are discussing an important topic about the potential benefits and dangers of various types of exercise in patients with chronic kidney disease. The molecular mechanisms of redox regulation that are involved in the stimulation and damage of muscle tissue and may play an important role in the prevention of muscular dystrophy in chronic kidney disease are considered.
The review is interesting and highlights important practical aspects, therefore it is recommended for publication. Please, correct some comments.
Comments:
Page 9 Authors write: «Proteins are the most susceptible to oxidative modifications, with the more common type of oxidation modification being carbonylation…» It is also necessary to mention other redox modifications that play an important role, in particular, S-glutathionylation.
In Figure 4, it is necessary to indicate the effect of exercise on the redox functiona and redox balance of cells, how exercises can stimulate antioxidant protection, and how it affects the pathways that are realized in CDK. It's not very clear what the arrow from Exercise to CKD means.
It is necessary to decipher the abbreviations - FAPs, PA.

Author Response

(The authors gave the same response as above.)
